In silico evaluation of molecular virus–virus interactions taking place between Cotton leaf curl Kokhran virus- Burewala strain and Tomato leaf curl New Delhi virus

Ali Nida Fatima 1
Paracha Rehan Zafar rehan@rcms.nust.edu.pk 2
Tahir Muhammad muhammad.tahir@asab.nust.edu.pk 1
1 Department of Plant Biotechnology, Atta-ur-Rahman School of Applied Biosciences (ASAB), National University of Sciences and Technology , Islamabad , Federal , Pakistan
2 Research Center for Modeling and Simulation (RCMS), National University of Sciences and Technology , Islamabad , Federal , Pakistan
Okpala Charles
Electronic publication date: 2021 Oct 19
Publication date: 2021
Volume: 9
Electronic Location ID: e12018
Received 2021 Mar 17; Accepted 2021 Jul 29
Copyright: ©2021 Ali et al.
Copyright year: 2021
Copyright holder: Ali et al.
License: This is an open access article distributed under the terms of the Creative Commons Attribution License, which permits unrestricted use, distribution, reproduction and adaptation in any medium and for any purpose provided that it is properly attributed. For attribution, the original author(s), title, publication source (PeerJ) and either DOI or URL of the article must be cited.
License URL: https://creativecommons.org/licenses/by/4.0/

Keywords: Virus–virus interactions, Cotton leaf curl disease, CLCuKoV-Bur, ToLCNDV, MD simulations, Protein protein interactions, Antagonistic interactions, Super infection exclusion

Funding: Atta-ur-Rahman School of Applied Biosciences (ASAB) National University of Sciences and Technology (NUST) The project was funded by Atta-ur-Rahman School of Applied Biosciences (ASAB), National University of Sciences and Technology (NUST). The funders had no role in study design, data collection and analysis, decision to publish, or preparation of the manuscript.

==============================
Background

Cotton leaf curl disease (CLCuD) is a disease of cotton caused by begomoviruses, leading to a drastic loss in the annual yield of the crop. Pakistan has suffered two epidemics of this disease leading to the loss of billions in annual exports. The speculation that a third epidemic of CLCuD may result as consequence of the frequent occurrence of Tomato leaf curl New Delhi virus (ToLCNDV) and Cotton leaf curl Kokhran Virus-Burewala Strain (CLCuKoV-Bu) in CLCuD infected samples, demand that the interactions taking between the two viruses be properly evaluated. This study is designed to assess virus-virus interactions at the molecular level and determine the type of co-infection taking place.

Methods

Based on the amino acid sequences of the gene products of both CLCuKoV-Bu and ToLCNDV, protein structures were generated using different software, i.e., MODELLER, I-TASSER, QUARKS, LOMETS and RAPTORX. A consensus model for each protein was selected after model quality assessment using ERRAT, QMEANDisCo, PROCHECK Z-Score and Ramachandran plot analysis. The active and passive residues in the protein structures were identified using the CPORT server. Protein–Protein Docking was done using the HADDOCK webserver, and 169 Protein–Protein Interaction (PPIs) were performed between the proteins of the two viruses. The docked complexes were submitted to the PRODIGY server to identify the interacting residues between the complexes. The strongest interactions were determined based on the HADDOCK Score, Desolvation energy, Van der Waals Energy, Restraint Violation Energy, Electrostatic Energy, Buried Surface Area and Restraint Violation Energy, Binding Affinity and Dissociation constant (Kd). A total of 50 ns Molecular Dynamic simulations were performed on complexes that exhibited the strongest affinity in order to validate the stability of the complexes, and to remove any steric hindrances that may exist within the structures.

Results

Our results indicate significant interactions taking place between the proteins of the two viruses. Out of all the interactions, the strongest were observed between the Replication Initiation protein (Rep) of CLCuKoV-Bu with the Movement protein (MP), Nuclear Shuttle Protein (NSP) of ToLCNDV (DNA-B), while the weakest were seen between the Replication Enhancer protein (REn) of CLCuKoV-Bu with the REn protein of ToLCNDV. The residues identified to be taking a part in interaction belonged to domains having a pivotal role in the viral life cycle and pathogenicity. It maybe deduced that the two viruses exhibit antagonistic behavior towards each other, and the type of infection may be categorised as a type of Super Infection Exclusion (SIE) or homologous interference. However, further experimentation, in the form of transient expression analysis, is needed to confirm the nature of these interactions and increase our understanding of the direct interactions taking place between two viruses.

Introduction

Cotton leaf curl disease (CLCuD) is a viral disease and notable menace to cotton productivity. CLCuD has been a serious threat to cotton production in Pakistan and India, leading to severe economic losses (Mansoor et al., 2003a; Mansoor et al., 2003b; Zubair et al., 2017). Vein thickening, leaf curling, and formation of small leaf-like enations on the lower surface of the leaf are the characterizing features of this disease (Padidam, Beachy & Fauquet, 1995). The viral agents associated with this disease are known as CLCuD associated begomoviruses (CABs), which belong to the family Geminiviridae, and genus Begomovirus (Mansoor et al., 1999). These viruses are transmitted by the vector whitefly (Bemisia tabaci) (Briddon & Markham, 2000). Geminiviruses are phytopathogenic viruses having a wide variety of hosts. They consist of circular, single-stranded DNA (ssDNA) genomes that are packed into icosahedral twinned particles, a feature unique to this family. Based on their host range, the vector of transmission, and genome arrangement these viruses have been arranged into nine genera (Varsani et al., 2017). Among them, the genus begomovirus is the most prevalent and diverse (Fauquet et al., 2008). The genome of begomoviruses can be monopartite, consisting of one DNA-A component or bipartite, comprising of two components, DNA-A and DNA-B, each having a size of 2.5 to 3.0 kb (Accotto et al., 1993). The genome of monopartite begomoviruses is small and consists of 6 to 8 Open reading frames (ORFS). The DNA-A encodes for the Coat protein (CP) and V2/AV2 protein in the virion sense strand, while the Replication initiation protein (Rep), Transcription Activation Protein (TrAP), Replication Enhancer Protein (REn), and the C4/AC4 proteins are encoded by a complementary sense strand (Hull, 2014). Some begomoviruses are occassionally associated with a symptom modifying DNA betasatellite (Family Tolecusatellitidae, Genus Betasatellite). These satellites are typically 1,350 bp in size, incorporating an adenine-rich region, the satellite conserved region (SCR) of a predicted hairpin structure that contains a loop region along with a single ORF that is named βC1, coded by the complementary strand (Briddon & Stanley, 2006). In the case of bipartite begomovirus, in addition to DNA-A, a DNA-B component is also present that encodes for the Movement protein (MP) and Nuclear Shuttle Protein (NSP). Viral proteins perform various functions inside the host cell. Rep is a multifunctional protein that carries out rolling circle replication (RCR) as it is involved in ATPase activity (Desbiez et al., 1995); it performs gene modulation and interacts with viral and host factors including REn, the Proliferating Cell Nuclear Antigen (PCNA), and Retinoblastoma-related Protein (pRBR) (Pradhan, Vu & Mukherjee, 2017). It is the most conserved geminiviral protein that has a critical role in the assembly of the viral replisome, a complex comprising of host and viral protein factors forming a DNA replication fork on the viral DNA (Ruhel & Chakraborty, 2019). TrAP creates a favorable cellular environment for viral replication, it plays a key role in viral pathogenicity, induces the transactivation of the host as well as viral genes (Haley et al., 1992), and causes suppression of gene silencing (Babu, Manoharan & Pandi, 2018). REn is required for the optimal replication of the viruses, interacts with Rep, and enhances the ATPase activity of this protein (Pasumarthy, Choudhury & Mukherjee, 2010). REn undergoes homo-oligomerization as it interacts with PCNA and pRBR, two host proteins that have a major role in host-cell cycle (Settlage, See & Hanley-Bowdoin, 2005). C4 protein consists of a highly conserved N-terminal myristoylation motif that is an irreversible post-translational modification in the protein structure, this domain functions as a RNA silencer in the cell (Fondong, 2013). CP encapsulates the ssDNA of the virus, protecting the nucleic acid and aiding its transmission both within and between plants (Frischmuth & Stanley, 1998). V2 protein suppresses RNA silencing and plays a role in viral infectivity and spread (Rojas et al., 2001). MP carries out cell-to-cell and long-distance movement of the virus while NSP transports the viral genomes from the nucleus to the cytoplasm and then back to the nucleus. MP guides the NSP-DNA complexes, as it modifies the plasmodesmata in the host cell and carries out trafficking of this complex from cell to cell. βC1 is involved in post-transcriptional gene silencing; it is a suppressor of transcriptional gene silencing and interferes with the metabolic and signaling pathways of the host cell (Brown et al., 2015). Details of begomoviral genome and the domain organization of different proteins are provided in Fig. 1.

Figure 1 Begomovirus genome and domain organisation of different viral proteins.

Monopartite genome is represented by CLCuKoV-Bu and Bipartite genome is represented by ToLCNDV. Domain organisation of different proteins consists of: (A) Replication Initiation Protein (Rep) DNA Nicking Domain (amino acids 1–120), DNA Binding Domain (amino acids 1–130) and the Oligomerization domains (amino acids 120–182). (B) Transcription Activation Protein (TrAP) (this protein maybe truncated (∼35 aa) in case of CLCuKoV-Bu or complete (∼134 aa) in case of ToLCNDV. The first 35 aa are similar are form similar domains, in both the cases, the protein consists of a basic region along with a potential nuclear localization sequence (NLS region; amino acids 17–31), Zinc finger region (ZFD; amino acids 33–55) Programmed cell death (PCD region; amino acids 20–55). (C) Replication Enhancer Protein (REn) consists of pRBR binding domains (pPBR-A; amino acids 1–13) (pPBR-B; amino acids 125–134), the PCNA binding domain , REn oligomerization domain (amino acid 28–128), Rep interacting domain (amino acid 28–128) along with three hydrophobic clusters (amino acids 18–28, 60–70, 80–110). (D) C4/AC4 protein of monopartite and bipartite viruses vary in size, the protein generally consists of N-terminal myristoylation motif (amino acids 1–18), sadenosyl methionine synthetase interacting domain (amino acid 13), SHAGGY-like kinase interacting domain (amino acids 30–56) and nuclear export sequence (NES; amino acids 65–81) (E) coat protein (CP) consists of the Nuclear Localization Sequences (amino acids 1–54, 100–127, 201–258) DNA binding domain (amino acid 1–121) cell wall targeting domain (CW; amino acids 100–127). (F) V2/AV2 protein consists of Putative Protein Kinase C (PKC) phosphorylation motif (amino acids 40–42), WCCH domain (amino acids 79–94) in the protein. (G) Movement protein (MP) consists of a pilot domain (amino acids 1–49), an anchor domain (amino acids 117-160) and an oligomerization domain amino (acids 161–250); (H) the nuclear shuttle protein (NSP) consists of an NLS-A (amino acids 21–42), NLS-B (amino acids 81–96), a DNA binding region (amino acids 39-109), MP interacting domain (amino acids 200–254), Nuclear Export Signal (NES; (amino acids 112 184–194),) and AtNSI interacting domain (amino acids 109–184). (I) CLCuMB βC1 protein consists of a myristoylation like motif (amino acids 101–108)

The Indo-Pak subcontinent has observed two CLCuD epidemics in the recent past. The first being the “Multan Epidemic”, occurring from 1989–1999 (Sattar et al., 2013) that led to a country wide spread of the disease, resulting in 80% yield loss causing a shortfall of US $5 billion to Pakistan’s economy (Briddon & Markham, 2000). In 2001, a recombinant begomovirus, the Cotton leaf curl Kokhran virus-Burewala strain (CLCuKoV-Bu) was observed near the town of Burewala in Punjab, Pakistan (Shahid et al., 2003). The unique feature of this new recombinant virus was that it possessed a truncated TrAP protein, consisting of only 35 amino acids (Amrao et al., 2010). Since the breakdown of resistance against CLCuD in 2001/2002 only the CLCuKoV-Bu, a monopartite begomovirus, in association with a recombinant form of Cotton leaf curl Multan beta-satellite (CLCuMuB) (Haider et al., 2006) has been associated with the disease. This spread of infection gave way to the second epidemic, which was observed from 2002 to 2013–2014. This was referred to as the “Burewala Epidemic” (Sattar et al., 2013). Until the Burewala endemic, only monopartite begomoviruses, in association with DNA satellites had been associated with the disease but the existence of bipartite begomovirus, the Tomato leaf curl New Delhi virus (ToLCNDV) in the CLCuD affected samples of the Cotton plant. (Zaidi et al., 2016) has raised concerns. This has led to the conjecture that a third epidemic maybe possible owing to the dual infection of these two begomoviruses, in the disease-affected host plants (Sattar et al., 2017). During dual or mixed infection of viruses, the interaction may range from synergism to neutralism to antagonism (Moreno & López-Moya, 2020). Virus-virus interactions (VVIs), defined as the measurable differences taking place in the plant during infection of one virus, accompanied by the concurrent or prior infection by a different viral strain or viral species. VVIs may be categorized as direct or indirect, based on the nature in which they occur. Direct VVIs take place between viral gene products of co-infecting viruses, while the indirect VVIs are observed due to changes in the host environment, these are also termed indirect environment interactions (DaPalma et al., 2010). The interaction taking place between the two begomoviruses under study are yet to be determined. This study is designed to understand the direct interactions taking place between the proteins of CLCuKoV-Bu and ToLCNDV. To our knowledge, this is the first time such a study has been designed. VVIs were observed in the form of electrostatic interactions, detected after performing molecular docking and undertaking the binding affinities and dissociation energies of the docked structures. By mapping out the interacting residues, the domains taking part in the interaction were highlighted. Based on our results, it is hypothesized that although concerns exist that dual infection could lead to a further epidemic of CLCuD, the opposite may also be true.

Materials & Methods

Sequence retrieval and PSI-BLAST analysis

Gene sequences for CLCuKoV-Bu and ToLCNDV were retrieved from Uniprot (http://www.uniprot.org). For CLCuKoV-Bu, the sequences of Rep (A0A0S2MSV0), TrAP (A0A0S2MST9), Ren (A0A0S2MSU6), C4 (A0A2K8HNC7), CP (A0A0S2MSU9), and V2 (A0A0S2MSV4) were retrieved. Along with the sequence for ßC1 (A0A0K2SU38) protein from the associating beta-satellite, CLCuMuB. For ToLCNDV, the sequences for Rep (A0A565D4R1), TrAP (A7WPF3), Ren (A6PYE1), AC4 (A7U6D0), CP (A0A2Z2GPB3), and AV2 (A0A3G2KQ59) proteins from DNA-A, and MP (E6N191) and NSP (A0A2P2CKV4) from DNA-B were obtained. These sequences were submitted to NCBI provided PSI-BLAST (https://www.ncbi.nlm.nih.gov/BLAST/) to identify homologous protein structures present in Protein Data Bank (PDB) (http://www.rcsb.org/pdb).

Protein structure prediction

Three different approaches were used for predicting the structure of the proteins of the two viruses. If the sequence identity for the viral gene products was greater than 75% then a homology-based approach was adopted, if it was less than 75% then threading-based and ab initio approaches were used for protein structure prediction. Homology modeling based structure prediction was performed using MODELLER 9.24 (Webb & Sali, 2014) where the homologous proteins were retrieved from PDB and were used as a template for modeling. Threading based approach was employed by submitting the amino acid sequences of the proteins, in FASTA format to I-TASSER (Iterative Threading ASSEmbly Refinement) webserver (Yang & Zhang, 2015) (http://zhanglab.ccmb.med.umich.edu/I-TASSER/), LOMETS version 3 webserver (LOcal MEta-Threading Server) (Wu & Zhang, 2007) (https://zhanglab.ccmb.med.umich.edu/LOMETS/), and RaptorX webserver (Källberg et al., 2012) (http://raptorx.uchicago.edu). I-TASSER is one of the most widely used online servers that is used for protein structure prediction and annotation of the structure-based functions. The server generates 3D models by matching the biological insights of the target proteins with the structures of known proteins present in biological databases (Roy, Kucukural & Zhang, 2010). LOMETS software is valuable in predicting the tertiary structure of the proteins. In this software, a 3D model is generated based on a consensus, devised by taking into consideration a number of models predicted by nine individual threading servers (Wu & Zhang, 2007). RaptorX program is useful in predicting protein models when no close template exists (Peng & Xu, 2009; Peng & Xu, 2010). Ab initio approach was performed by submitting gene sequences comprising of 200 or fewer amino acids to the QUARK webserver (Xu & Zhang, 2013) (https://zhanglab.ccmb.med.umich.edu/QUARK/).

Model quality assessment

The quality of the predicted protein models was evaluated using different methods. Stereochemical quality and structural rationality were evaluated using the ERRAT (Colovos & Yeates, 1993) (https://servicesn.mbi.ucla. edu/ERRAT/), global model quality assessment was done through QMEANDisCo (Qualitative Model Energy Analysis Distance Constraint) (Benkert, Künzli & Schwede, 2009) (https://swissmodel.expasy.org/qmean/. QMEANDisCo is a composite program that provides global and local quality estimates for the protein model. The distant constraint (DisCo) feature evaluates the pairwise distance that exists within the residues in the model and assesses the multitude of constraints that may exist within the structure, based on the information from experimentally determined protein structures that are homologous to the protein model being assessed. It is calculated as the average score per residue and occurs in the range of 0 to 1 (Studer et al., 2020). ProSA-Web Z-Score (Wiederstein & Sippl, 2007) (https://prosa.services.came.sbg.ac.at/prosa.php) is a diagnostic tool that is based on the statistical analysis of all the available protein structures. This program relates to the overall quality score of the protein structure in the form of a plot comprising experimentally determined protein structures that are present in the PDB. The overall quality of the protein models was estimated using global checks on the model’s stereochemistry, geometry, and other structural properties. This was done utilizing Ramachandran Plot, obtained through PROCHECK (Laskowski et al., 1993).

Intra-viral and inter-viral protein–protein interactions (PPI)

Intra-viral PPI, (taking place between the proteins of the same viruses) and Inter-viral PPI (taking place between the proteins of two different viruses) were performed to evaluate whether a stronger interaction is observed within the proteins of the same virus, or between the proteins of different viruses. PPIs of the proteins of CLCuKoV-Bu and ToLCNDV encoded proteins were studied using the High Ambiguity Driven bio-molecular DOCKing (HADDOCK2.2) web portal (De Vries, Van Dijk, & Bonvin, 2010; Van Zundert et al., 2016) (https://milou.science.uu.nl/services/HADDOCK2.2/). HADDOCK is a widely used docking tool that is based on information-driven docking protocol. It requires the generation of an Ambiguous Interaction Restraints (AIRs) file for a manual run. The prerequisite to the AIRs file is to define the active and passive residues present at the interface of each molecule. These residues are defined using experimental data or through bioinformatic predictors like CPORT (Consensus Prediction Of Residues in Transient complexes) (De Vries & Bonvin, 2011) (http://haddock.science.uu.nl/services/CPORT/). The HADDOCK server distinguishes between two types of residues. The active residues must be present at the interface between the two molecules and maybe in contact with the active or passive residues of the other molecule, if not, an energy penalty would be paid. The passive residues, on the contrary, may be present at the interface but are not penalized if found otherwise (Hinchliffe & Harwood, 2019). A HADDOCK run is set up by providing two proteins whose interaction is to be evaluated, in PDB format to the server. The list of active and passive residues is also provided as input data for each protein. The docking protocol follows three steps, rigid body minimization, semi-flexible refinement, and the final refinement of the models that take place in an explicit solvent. 1,000 models are produced by default, as a result of the initial rigid-body docking. The default number of models is reduced to 200 after semi-flexible refinement and final refinement in water. These final water-refined models are clustered and ranked based on their respective HADDOCK score. This clustering is done based on pairwise Root Mean Square Deviations (RMSD), with the default value of RMSD being set to a cutoff of 7.5 Å, and the minimum number of structures to define a cluster is four (by default) (Saponaro et al., 2020). Detailed statistics that represent the average values of the HADDOCK score along with other standard energies including Van der Waals, Desolvation energy, Restraint violation energy, Intermolecular binding energy, non-bonded interaction energy, buried surface area, and Z-score were calculated for the high scoring complexes in each cluster.

Determination of interacting residues

The protein–protein complexes were selected based on their HADDOCK scores and associative energies. The binding energies and the intermolecular contacts present within the complexes were determined by submitting the top scoring complex, for each interaction, to PRODIGY (PROtein binDIng enerGY predict) webserver (Vangone & Bonvin, 2015). It is a contact-based predictor of binding affinity, which is described in the form of Gibbs free energy (ΔG, kcal mol-1) and dissociation constant (Kd, M). Interatomic contacts (ICs) at a distance threshold of about 5.5 Å within the protein–protein complex are counted and classified based on the Polar/Non-Polar/Charged nature of the interacting amino acids. The server also provides the number of residues in the Non-Interacting Surface (NIS) of the protein (Vangone & Bonvin, 2017). Both ICs data, along with the information of NIS properties influence the binding affinities of the complexes (Kastritis et al., 2011). The structures were visualized through PyMol (Schrödinger, 2015) (https://pymol.org/). By performing sequence analysis through Jalview software (Waterhouse et al., 2009) (https://www.jalview.org/) the interacting residues determined through PRODIGY server were highlighted, and the domains these residues belonged to, were identified.

Molecular dynamics (MD) simulations of docked protein complexes

Molecular dynamics (MD) simulations provide identification of the conformational changes and the evaluation of the molecular interactions that are present within the complex assembly (Wang et al., 2020). MD simulations were performed for the viral complexes; Complex 1: Rep (CLCuKoV-Bu) - MP (ToLCNDV); Complex 2: Rep (CLCuKoV-Bu) - NSP (ToLCNDV) that exhibited the greatest affinity, lowest binding energies, and high dissociation energies among all the interactions. MD simulations were performed by using Groningen Machine for Chemical Simulations (GROMACS) v.5.1 (Abraham et al., 2015) . It is a reliable tool that is used to evaluate different biological models within realistic cellular environments. Default parameters were used for the simulations. All the systems were solvated in a cubic water box, with the complexes being placed at least 1.0nm from the edge of the box. The Optimized Potential for Liquid Simulation (OPLS) force field (Robertson, Tirado-Rives & Jorgensen, 2015) was used. The physiological pH of the systems was maintained using the Single Point Charge Extended (SPCE) water model (Wu, HL & Voth, 2006), and to neutralize the systems, Na+ or Cl− ions were added. The steepest descent approach was used for energy minimization for 50,000 cycles. The systems were then equilibrated for a constant number of particles, volume, temperate (NVT), and a constant number of particles pressure, temperature (NPT). Following the equilibrations, 50 ns simulation was performed for the protein complexes.

Results

Primary data Retrieval

Sequences for the genes of CLCuKoV-Bu and ToLCNDV were retrieved from Uniprot, and PSI-BLAST analysis were performed to find homologous protein sequences. The information is provided in Table 1.

Table 1 Sequence Retrieval and PSI-BLAST result for Genes for DNA-A CLCuKoV-Bur, βC1 gene of CLCuMB and DNA-A and DNA- B of ToLCNDV.

Virus	Uniprot accession no.	PSI- BLAST result	
			PDB ID	Total score	Query cover	E value	Percentage identity	
CLCuKoV-Bur	Rep	A0A0S2MSV0	1L2M	206	31%	2e−66	78.45%	
	TrAP	A0A0S2MST9	6DDT	24.6	48%	52.94%	52.94%	
	REn	A0A0S2MSU6	2XF4	32.7	44%	0.1	30.16%	
	C4	A0A2K8HNC7	2CKF	16.2	26%	0.74	23.68%	
	CP	A0A0S2MSU9	6F2S	394	83%	1e−140	82.79%	
	V2	A0A0S2MSV4	3FCX	27.7	31%	4.0	35.14%	
CLCuMB	βC1	A0A0K2SU38	6JWP	28.5	69%	2.6	28.05%	
ToLCNDV DNA-A	Rep	A0A565D4R1	1L2M	208	32%	3e−67	82.76%	
	TrAP	A7WPF3	6F12	30	39%	1.1	30.36%	
	REn	A6PYE1	4N0H	27.7	30%	6.4	35.56%	
	C4	A7U6D0	3H74	30.8	55%	0.05	37.5%	
	CP	A0A2Z2GPB3	6F2S	367	83%	7e−130	78.6%	
	V2	A0A3G2KQ59	1Q9J	27.7	46%	3.6	40.74%	
ToLCNDV DNA-B	NSP	A0A2P2CKV4	5E68	30.4	8%	1.8	52.17%	
	MP	E6N191	3N2O	32.3	23%	0.81	27.27%	

Protein structure prediction and assessment

3D structures of the proteins were generated through homology, threading and ab initio-based approaches, by using five different software. Homology modeling was performed using MODELLER. Through this approach, the structures of Rep and CP of CLCuKoV-Bu and ToLCNDV, which showed percentage sequence identity greater than 75% after PSI-BLAST analysis, were produced. In the case of ab initio modeling, done through QUARK, only the proteins comprising of less than 200 amino acids were modeled. These included the structures of TrAP, REn, C4, V2 proteins of CLCuKoV-Bu, βC1 protein of CLCuMuB, and TrAP, REn, C4, and V2 proteins of ToLCNDV. By using software like I-TASSER, LOMETS, and RAPTORX, structures for all the proteins of the two viruses were produced, by the threading-based approach.

Protein structure quality assessment

The protein structures generated through different approaches were evaluated using ERRAT, QMEAN, ProSA Z-Scores, and Ramachandran plot, and a consensus structure for each protein was selected based on the results of these assessment tools. The structures produced by Homology modeling were the least stable, while those generated by Ab initio and threading based approaches proved to be the most stable. QUARK and I-TASSER were the most efficient programs. The overall quality of the structures assessed by ERRAT was found to be greater than 80% for the models of all proteins of both the viruses, except for βC1 of CLCuMuB, which had a value of 74%. The QMEANDisCo score, of all the protein structures, exhibited structure stability with their scores occurring within the given range. The Ramachandran plot exhibited that the maximum number of residues in the favorable and allowed regions. The Z-score by ProSA-web exhibited the structures were reliable, with the energy conformation of the models lying within a range of scores that are usually found for similar-sized proteins. Detailed results of structure evaluation are presented in Table S1. The structure for the consensus model of each protein is provided in Fig. 2.

Figure 2 Consensus models of the proteins of CLCuKoV-Bu and ToLCNDV generated through different approaches are presented.

Intra-viral and inter-viral PPIs

PPIs were performed to access the intermolecular interactions that may be taking place between the proteins of the two viruses. This is a crucial step in accessing the domains taking part in the interaction and identifying the proteins showing the most affinity for the proteins of the other virus. CPORT analysis was performed to identify the active and passive residues of the proteins, provided in Table S2.

The PPIs performed using HADDOCK2.2 webserver showed affinity among the proteins of the two viruses. In total 169 interactions were performed, with 56 interactions taking place between the proteins of CLCuKoV-Bu and ToLCNDV, 49 between the proteins of CLCuKoV-Bu and CLCuKoV-Bu, and 64 between the proteins of ToLCNDV and ToLCNDV. The inter-viral PPIs, between CLCuKoV-Bu and ToLCNDV were the strongest based on the HADDOCK scores and corresponding energy values. Rep and CP of CLCuKoV-Bu and MP and NSP of ToLCNDV were most actively involved in the interactions. Among all the interactions, the strongest that were observed involved the Rep (CLCuKoV-Bu) protein. As was seen in the interaction of Rep (CLCuKoV-Bu) – MP (ToLCNDV), Rep (CLCuKoV-Bu) – NSP (ToLCNDV) and Rep (CLCuKoV-Bu) - Rep (ToLCNDV). Another significantly strong interaction was observed between REn (CLCuKoV-Bu) - NSP (ToLCNDV). The docked structures were evaluated based on their RMSD values and energies like, desolvation energy, Van der Waals energy, electrostatic energy, restraint violation energy, buried surface area and the Z- score, that is based on the probability of the results among the various clusters. The details of these interactions, HADDOCK scores and energy values of the complexes are provided in Fig. 3. Detailed results of intra and inter viral PPIs are presented in Table S3 while some of the stronger interactions based on the HADDOCK scores and corresponding energy values are presented in Table 2.

Figure 3 HADDOCK provided scores for some of the strongest observed interactions.

(A) Scores of Clusters 1-7 for PPI of Rep (CLCuKoV-Bu) with MP (ToLCNDV): The complex had the HADDOCK scores of −279.8, −259.2, −238.3, −237.5, −235.9, −233.1, −216.9; desolvation energy in (kcal/mol) −125.7, −110.9, −131.3, −122.1 , −139.1, −128.2; Van Der Waals energy values of −120.5, −125, −114.4, −110.1, −106, −106.8, −83.5; Restraints Violation Energy values of 523, 514, 586.2, 605.1, 534.1, 530.1, 605.1; Z-Score of −2, −1.2, −0.3,−0.3, −1.2,−0.1, 0.6; cluster size of 47,7,9,26,14,14,5; RMSD (Å) values of 10.6, 13.4, 1.8, 6.2, 4.7, 4.1, 11.8; buried surface area of 3939, 3636, 3470, 3417, 3034, 3091, 3478 and Electrostatic Energy of −429.9, −373.3, −255.8, −329.4, −220.5, −255.4, −240. (B) Scores of Clusters 1–7 for PPI of Rep (CLCuKoV-Bu) with NSP (ToLCNDV): the complex had the HADDOCK scores of -241.5, -223.4, -221.7, -217.8, −211.9, −211.4, −198.3; desolvation energy in (kcal/mol) −144, −144, −150.3, −124.3, −140.8, −157.1, −140.4; Van Der Waals energy values of −130.1, 115, −107.1, −114.5, −107.5, −108.7, −101.4; Restraints Violation Energy values of 721.5, 789.7, 906.7, 877.9, 736.3, 933, 758.6; Z-Score of −2.1, −0.8, −0.7, −0.5, −0.1, 0, 0.9; Cluster size of 24, 17, 35, 8, 13, 10, 6; RMSD (Å) values of 13.1, 14.9, 18.1, 2.1,18.5,6.3,14.3 ; Buried surface area of 3733.4, 3345.7, 2960.4, 3677.7, 2983.7, 2939.2, 3116.5 and electrostatic energy of −197.9, −216.8, −275.1, −333.8, −185.8, −194.3, −161.1. (C) Scores of Clusters 1-7 for the PPI of REn (CLCuKoV-Bu) –NSP (ToLCNDV): the complex had the HADDOCK scores of −217.8, −198, −85, −50, −49.9, −35.4, −5.1; Desolvation energy in (kcal/mol), −124.3, −140, −46.6, −72.4, −78.8, −72.4, −65.3; Van Der waals energy values −114.5, −101.5, −115.7, −99.5, −100, −70.8, −236; Restraints Violation Energy values of 1877, 1646, 1703. 1419, 1588, 1724, 1654; Z-Score of 0.1, 0.5, −2.6, −0.6, −0.6, 0.3, 0.5; Cluster size of 8, 6, 16, 9, 7, 8, 5; RMSD (Å) values of 2.1, 14.3, 7, 10.8, 12, 16.3, 15.6; Buried surface area of 2889.2, 2732.7, 2756.8, 2188.6, 2513.2, 2409.6, 2377.6 and electrostatic energy of −220, −172.9, −186.4, −163.8, −175.8, −170.8, −161.3. (D) Scores of Clusters 1-7 for the PPI of Rep (CLCuKoV-Bu) with Rep (ToLCNDV): the complex had the HADDOCK scores of −199.9, −136, −201, −162.8, −124.4, −186.7, −95; Desolvation energy in (kcal/mol) −159, −165.5, −133.8, −148.6, −158.6, −157.2, −152.8; Van Der waals energy values −138.9, −145.4, −121.3, −94.6, −114.3, −117.9, −102; Restraints Violation Energy values of 1541, 1817.2, 1869.5, 1390, 1779.3, 1510.2, 2001.9; Z-Score of −1.4, 0.5, −1.4, −0.3, 0.8, −1, 1.7; Cluster size of 10, 33, 11, 8, 6, 6, 5; RMSD (Å) values of 25.3, 9.2, 14.6, 4.4, 19.6, 1, 25.1; Buried surface area of 33825.9, 3552.4, 3927.4, 2922.9, 3310.6, 3678.1, 3039.4 and Electrostatic Energy of −215.3, −313.1, −450.1, −293.2, −146.9, −313.1, −202.1.

Overall, all the observed interactions exhibited strong binding affinity, based on the recorded energy values, only some interactions with the REn protein of CLCuKoV-Bu seemed to be weak, exhibiting low dissociation energy and low affinity values. Details of the energy values for the observed interactions are provided in Table S4 and binding affinity and dissociation constant (Kd) values of the docked structures, calculated by the PRODIGY webserver are provided in Table S5.

Residue identification

The amino acids residues taking part in the interaction were determined by submitting the docked complexes to PRODIGY server. The maximum number of interacting residues among all the strong PPIs belonged to Polar-Apolar ICs, while the minimum number belonged to Charged-Charged ICs. Details of charges of the interacting residues are given in Table S6.

Table 2 Strongest interactions observed in case of inter-viral and intra-viral infections.

Virus/Satellite 1	Protein	Virus/Satellite 2	Protein	HDK1 Score a.u.	RMSD Å	DE2	VDW3 Energy	ESE4	BSA5	
CLCuKoV-Bu	Rep	CLCuKoV-Bu	Rep	−198	25.2	−131	−107	−230	2892	
CLCuKoV-Bu	Rep	CLCuKoV-Bu	CP	−193.7	21	−112	−85	−184	2481	
CLCuKoV-Bu	Rep	CLCuKoV-Bu	V2	−153.6	9.9	−74.7	−88.6	−294	2459	
CLCuKoV-Bu	Rep	ToLCNDV	Rep	−201	14.6	−159	−138.9	−450	3927	
CLCuKoV-Bu	Rep	ToLCNDV	TrAP	−195.1	1.9	−129	−134	−177	3795	
CLCuKoV-Bu	Rep	ToLCNDV	C4	−196	1	−104	−93	−232	2407	
CLCuKoV-Bu	Rep	ToLCNDV	CP	−192.4	8.9	−158	−100	−169	2903	
CLCuKoV-Bu	Rep	ToLCNDV	MP	−279.8	10.6	−125	−120	−429	3938	
CLCuKoV-Bu	Rep	ToLCNDV	NSP	−241.5	13.1	−144	−130.1	−197	3733	
CLCuKoV-Bu	TrAP	ToLCNDV	TrAP	−103	2.5	−41.9	−105.2	−220	2990	
CLCuKoV-Bu	TrAP	ToLCNDV	MP	−134.9	0.7	−23.5	−86.8	−256	2647	
CLCuKoV-Bu	TrAP	ToLCNDV	NSP	−129.6	0.7	−44	−85.3	−314	2235	
CLCuKoV-Bu	REn	ToLCNDV	NSP	−217.8	2.1	−124	−114	−333	3677	
CLCuKoV-Bu	C4	ToLCNDV	MP	−160.9	1	−50	−120.1	−385	3762	
CLCuKoV-Bu	CP	ToLCNDV	TrAP	−185.3	0.7	−95.4	−131.5	−345	4146	
CLCuKoV-Bu	CP	ToLCNDV	CP	−163.4	0.6	−112	−127	−164	3991	
CLCuKoV-Bu	CP	ToLCNDV	MP	−166.9	2.1	−51.5	106.1	−455	3553	
CLCuKoV-Bu	V2	ToLCNDV	MP	−199.4	0.9	−63.5	−106	−404	3092	
CLCuMuB	ßC1	ToLCNDV	MP	−199.7	0.8	−60	−115.1	−500	3447	
ToLCNDV	TrAP	ToLCNDV	Rep	−191.8	0.8	−137	−149	−341	4396	
ToLCNDV	TrAP	ToLCNDV	TrAP	−120	2.5	−51.9	−110.2	−320	3990	
ToLCNDV	AC4	ToLCNDV	AC4	−160.6	1	−80	−100	−268	2225	
ToLCNDV	TrAP	ToLCNDV	CP	−149.3	8.8	−51.4	−121	−430	4078	
Notes.

1 HADDOCK

2 Desolvation Energy

3 Van Der Waals Energy

4 Electrostatic Energy

5 Buried surface Area

6 Restraint Violation Energy

Based on the knowledge of these interacting residues, the domains involved in the interaction were identified using Jalview software. For the two begomoviruses, similar domains were identified to take part in the interactions. This was deduced after taking into account all 169 interactions and identifying the interacting amino acid residues. For the Rep (CLCuKoV-Bu) and Rep (ToLCNDV), the DNA Binding and the Helicase/ATPase domains were involved in interactions. The majority of the interacting residues belong to the helicase domain, while only one amino acid residue from the DNA binding domain seemed to be involved, these residues are presented in Fig. 4A. TrAP (CLCuKoV-Bu) was truncated and only the amino acids from the nuclear localization sequence (NLS) were involved in interactions, while in case of TrAP (ToLCNDV) the Zinc finger-like motif, the programmed cell death (PCD) domain and the acidic region at the C terminal were involved in the interactions: Fig. 4B.

Figure 4 Identification of the amino acid residues taking part in interaction in Rep, TrAP and REn proteins of both CLCuKoV-Bu and ToLCNDV.

(A) Rep protein, CLCuKoV-Bu (top, 1–363 amino acids) and ToLCNDV (bottom, 1–358 amino acids). The Rep protein consists of DNA- Nicking domain 1–120, DNA Binding Domain (1–130), oligomerization domain (amino acids 120–182) at the N-terminal while the C terminal consists of ATPase/Helicase domain. The 39th amino acid from the Nicking/ DNA binding domain is involved in the interaction with the majority of the interacting residues occurring in the ATPase/Helicase domain. (B) TrAP protein, CLCuKoV-Bu (top, 1–35 amino acids) and ToLCNDV (bottom, 1–139 amino acids). The residues from the NLS domain (amino acids 20–30) are taking part in interaction for the TrAP of CLCuKoV-Bur while for the TrAP of ToLCNDV the residues from the NLS, ZFD (amino acids 33–55), and PCD (amino acids 20–55) domain were involved in interaction. (C) REn protein, CLCuKoV-Bu (top, 1–134 amino acids) and ToLCNDV (bottom, 1–136 amino acids). The interacting residues for REn protein of both the viruses occurs in the RBR (amino acids 1–8) domain, REn oligomerization domain (amino acids 28–95), PCNA binding domain (amino acids 7–95), SINAC1 domain (amino acids 1–80).

The REn protein of CLCuKoV-Bu and ToLCNDV, exhibited that both N and C terminals were involved in interactions. The interacting residues occurred in the Rep binding domain, PCNA binding domain, SINAC1 binding domain, and the retinoblastoma related protein binding regions: Fig. 4C. AC4 (ToLCNDV) was smaller in size as compared to C4 (CLCuKoV-Bu), however, in both the proteins, residues from the N-terminal myristoylation sequence, the S-adenyl methionine synthetase interacting residue and the SHAGGY-like Kinase interacting domain were involved in the interaction, Fig. 5A. The CP of both the begomoviruses had majority of the interacting amino acids belonging to the Nuclear Localization Signal (NLS) region, and the DNA binding domain present at the N-terminal. The other domains of this protein involved in interaction included the Cell Wall targeting motif (CW), the Nuclear Export Signal (NES) and the NLS domain present at the C terminal, Fig. 5B.

Figure 5 Identification of the amino acid residues taking part in interaction in C4, CP and V2 proteins of both CLCuKoV-Bu and ToLCNDV.

(A) C4 protein of CLCuKoV-Bu (top, 1–146 amino acids) and AC4 ToLCNDV (bottom, 1–58 amino acids). The interacting residues for C4/AC4 protein of the viruses occur in the N-terminal myristoylation sequence (1–8 amino acids), the S-adenyl methionine synthetase interacting residue (13th residue) and the SHAGGY-like Kinase interacting domain (51st residue). No residue from the nuclear export signal NES (65–81 amino acids) domain was involved. (B) CP of CLCuKoV-Bu (top, 1–256 amino acids) and ToLCNDV (bottom, 1–256) amino acids. The interacting residues occur in the NLS (1–54 amino acids) and the NES/CW (100-127 amino acids) domains. Some residues also occur in the NLS domain at the C terminal. (C) V2 protein of CLCuKoV-Bu (top, 1–118 amino acids) and AV2 ToLCNDV (bottom, 1–112 amino acids). The V2 protein of CLCuKoV-Bu has residues in the PKC phosphorylation motif (40–42 amino acids) and the WCCH domain (79–94 amino acids) while for AV2 of ToLCNDV only the residues in the WCCH domain took part in interaction.

V2 (CLCuKoV-Bu) showed participation of the Putative Protein Kinase C (PKC) domain and the WCCH domain in interaction, while for the AV2 (ToLCNDV), interacting residues were only present in the latter domain, and no participation was shown for the former, Fig. 5C.

The βC1 protein of the CLCuMuB had several residues involved in the interactions, ranging from the center to the C-terminal of this protein. Interacting residues also occurred in the myristoylation domain of this protein, Fig. 6A.

Figure 6 Identification of the amino acid residues taking part in interaction in βC1 of CLCuMuB and MP and NSP of ToLCNDV.

(A) β C1 protein of CLCuMuB The residues from the myristoylation like motif domain (101-108 amino acids) were involved in the interaction. (B) MP protein from DNA-B of ToLCNDV. The residues from the pilot domain (1–49 amino acids), Anchor domain (117–160 amino acids) and the Oligomerization domain (161–250 amino acids) to be involved in interaction. The residues from the NSP interacting domain may also be involved in the interactions. (C) NSP protein from DNA-B of ToLCNDV. The residues from the NLS (21–42 amino acids), DNA binding domain (39–109 amino acids), MP interacting domain (200–254 amino acids), and the AtNSI interacting domain (109–184 amino acids), no residues from the NES domain (184–194 amino acids) took part in the interactions.

MP (ToLCNDV-DNA B) had the strongest interaction with the proteins of CLCuKoV-Bu and it exhibited interacting residues from the Pilot, anchor and oligomerization domains. with the majority of the interacting residues belonging to the latter. As the NSP interacting domain also overlaps with the anchor and oligomerization domains, it may also be involved in the interactions, Fig. 6B.

The NSP (ToLCNDV- DNA B) had the interacting residues present in the NLS-A region while the residues from NLS-B did not take part in interactions. The residues from the initial parts of the DNA binding domain, the MP interacting domain and the AtNSI interacting domains were involved in the interactions, Fig. 6C.

Molecular dynamics simulations

To understand key structural details, in terms of stability, structural integrity, and compactness of the complexes of the interacting proteins, MD simulations were carried out. The HADDOCK server also conducts water-based simulations, therefore, to remove steric hindrances present within the structures of the complexes 50 ns simulations were performed. The Root Mean Square Deviation (RMSD) and Root Mean Square Fluctuation (RMSF) analyses provide information about structural stability and flexibility of the protein–protein complexes.

RMSD value for the average MD simulation conformation is used to quantitate how much the protein has changed or how much the conformations vary in the duration of the simulation. The results show that the maximum, minimum and average RMSD values for complex 1 are 0.65, 0.25, 0.45 and for complex 2 are 0.78, 0.2, 0.65, exhibiting structure stability of the molecules throughout the duration of the simulation. Fluctuation of the individual amino acid residues within the complexes can be calculated based on the RMSF value, obtained from the 50 ns MD simulation. The results show that the maximum, minimum and average RMSF values are 1.45, 0.06 and 0.75 for complex 1 and 1.25, 0.1 and 0.675 for complex 2. Low average RMSF values indicate that individual amino acids exhibit stability in the dynamic state of the protein during MD simulation. The Radius of gyration (Rg) was calculated to determine the compactness of the complexes. Rg provides the overall dimensions of the protein structure as it shows the mass-weighted root mean square distance of a collection of atoms from their common center of mass. Results show the average Rg value of 2.66 for complex 1 and 2.825 for complex 2 for a 50 ns simulation at 300 K. Hydrogen bonds are considered to play a vital role in the structural stability and molecular interaction of the proteins. The greater number of hydrogen bonds helps to maintain the stability of the structures. The hydrogen bond analysis was carried out through the VMD 1.9.4 program (Humphrey, Dalke & Schulten, 1996) for the MD trajectories.

The results obtained through these analyses signified that the interactions between the protein structures were stable and the docked complexes remained structurally sound throughout the duration of the simulation. Graphical representation of the results for complexes 1 and 2 are provided in Fig. 7.

Figure 7 50 ns MD simulation analysis for Complex 1 Rep (CLCuKoV-Bu) and MP (ToLCNDV) and Complex 2 Rep (CLCuKoV-Bu) and NSP (ToLCNDV).

(A) RMSD: complex 1 had the lower RMSD as compared to complex 2, average value of RMSD for complex 1 was 0.45 and 0.65 for complex 2. (B) RMSF: the average value of RMSF for complex 1 is 0.75 for complex 2 it is 0.675. (C) Radius of Gyration (Rg): the average value of Rg was 2.66 for complex 1 and 2.825 for complex 2 during the 50 ns simulation. (D) H-Bond Analysis: the complexes are stable due to high number of hydrogen bonds present.

Discussion

The coinfection of ToLCNDV and CLCuKoV-Bu have raised the speculation of a third epidemic of CLCuD (Sattar et al., 2017). Direct interactions taking between the proteins of CLCuKoV-Bu with the proteins of ToLCNDV have been observed, in order to hypothesise the nature of the interactions taking place between two begomoviruses. Mixed infections of ToLCNDV with several monopartite begomoviruses have been reported, and the virus is known to infect a wide range of plants belonging to various plant families, including Cucurbitaceae, Solanaceae, Malvaceae, Euphorbiaceae and Fabaceae (Zaidi et al., 2017). In case of such infections, each viral population changes the environment and fitness landscape inside the host, and the success of any virus depends not only on its adaption to the host but also on how it interacts with the other virus (Elena, Bernet & Carrasco, 2014). In this study, in silico models were generated to test the hypothesis that the coinfecting viruses interact with each other. Such models may help in understanding the pathogenesis and pathophysiology of a disease, which can lead to the development of control strategies (Barh et al., 2013). The involvement of host factors is omitted for simplicity purposes along with presuming the differences that may occur in localization and the timing of events. A limitation of this study is that the structure prediction of the viral proteins had to be primarily based on template-free methods, due to the lack of homologous protein structures available in the PDB (Chen & Weng, 2003; Van Zundert et al., 2016). Another limitation is the inability of the protein models to account for the high rates of mutation occurring in the viral protein sequences (Echave, Spielman & Wilke, 2016). Our results indicate a strong affinity between the proteins of the two begomoviruses, which may influence their rate of infection and spread within the host. From among the proteins of CLCuKoV-Bu and CLCuMuB, the strongest interactions that were observed involved the Rep protein, similarly for ToLCNDV the proteins of DNA-B were involved in the strongest interactions.

Rep of CLCuKoV-Bu exhibited a strong affinity for the proteins of the other virus with the ATPase/Helicase domain of Rep protein most frequently involved in the interactions. Rep is the most important protein in the viral life cycle. It is involved in a number of processes, ranging from the transcriptional activation and suppression of viral gene products, interacting with various factors in the host cell and ATPase/Helicase activity. The involvement of the ATPase domain of Rep in PPIs may hinder the RCR mechanism of viral replication that can reduce the number of viral agents in the host cell (Desbiez et al., 1995). Interaction of Rep with REn promotes accumulation of viral DNA (Settlage, See & Hanley-Bowdoin, 2005), and further interaction with CP suppresses the nicking and ligation activity of this protein (Malik et al., 2005). The strong binding of Rep with MP and NSP may cause hindrance in the formation of these complexes. The N and C terminal domains of Rep reported to be involved in PPIs are important for DNA recognition, binding and ATPase/Helicase activity (Chatterji et al., 2000). The REn protein exhibited a very strong interaction with the NSP of ToLCNDV (DNA-B). Although REn is capable of homo-oligomerization (Settlage, See & Hanley-Bowdoin, 2005) it exhibited very weak bonding with the REn of the different virus and a stronger interaction was observed between REn proteins of the same virus.

The TrAP of CLCuKoV-Bu comprising a coding capacity of 35 amino acids is considered truncated (Amrao et al., 2010). This protein suppresses post-transcriptional gene silencing (PTGS) in the host and modulates microRNA (miRNA) expression. These characters are retained in the complete (∼134 aa) as well as in the truncated (∼35 aa) TrAP proteins (Akbar et al., 2016). This protein is capable of undergoing self-interaction with the TrAP-TrAP complexes accumulating in the nucleus (Yang et al., 2007; Guerrero et al., 2020). The interactions observed between the TrAP (CLCuKoV-Bu) - TrAP (ToLCNDV) and TrAP (ToLCNDV) - TrAP (ToLCNDV), validate the self-interaction concept. A strong interaction of CP (CLCuKoV-Bu) was observed with TrAP (ToLCNDV). This is in validation with the work of Guerrero et al. (2020), which states that TrAP protein is involved in regulating the expression of CP protein during the viral lifecycle. C4 protein suppresses the post-transcriptional gene silencing (PTGS) by interacting with the NbSAMS2 host protein. The first 20 amino acid residues of the protein play a crucial role in the binding of this protein (Ismayil et al., 2018). The results of the current study show the first 23 amino acids residues, comprising the N-terminal myristoylation and S-adenosyl methionine synthetase regions (Fondong, 2019) to be involved in the PPIs.

The MP and NSP of DNA-B of ToLCNDV are involved in the cell to cell transport of the viral products and in the export of the Viral DNA from the nucleus (Zhang, Wege & Jeske, 2001), these functions are achieved by the interaction between MP and the NSP (Rojas et al., 1998). Our findings show these two proteins are actively involved in inter-viral interactions. As was observed in the interactions between the proteins of DNA-B of ToLCNDV with the βC1 (CLCuMuB), suggesting an association between the betasatellite and the bipartite begomovirus (Jyothsna et al., 2013). A significant interaction noted between βC1 (CLCuMuB) and MP (ToLCNDV), with a HADDOCK score of −199 and ΔG value of −15.8 (kcal mol−1), indicated that the protein–protein complex was stable, with the amino acids from the Myristoylation-like motif domain of βC1 and pilot domain, the anchor domain and oligomerization domains of MP were involved in the interaction.

The outcome of interactions between dual viral infections may range from synergism to neutralism to antagonism, with each having a direct impact on the health of the plant. In case of no change in the viral accumulation as compared to single virus infection, the state is referred to as neutralism, whereas if the accumulation of either one or both of the viral agent is enhanced it is called synergism and if the population of either one or both viral agents is reduced it is termed as antagonism (Moreno & López-Moya, 2020). The direct interactions between the gene products of the two similar begomoviruses show that key domains, playing monumental roles in the viral life cycle, from replication of the virus to its export from the nucleus to the spread to the other cells take part in the PPIs. These interactions are hypothesized to limit the growth of both CLCuKoV-Bu and ToLCNDV, limiting their spread and in effect reduce the virulence of the CLCuD. Therefore, this may be classified as an antagonistic interaction. This mechanism may be a type of cross-protection or homologous interference, that is an antagonistic interaction occurring between similar viruses (Syller & Grupa, 2016) Antagonistic interactions are relatively less reported as compared to synergistic interactions, because they result in disease attenuation in contrast to exacerbation and therefore may go unnoticed (Díaz-Muñoz, 2019). These findings are based on in-silico research and serve as the first step towards understanding the mechanism in which virus-virus interactions could be taking place. It is important to carryout in planta experimentation and evaluate these findings.

Conclusions

The present study aims to develop an understanding of the nature of coinfection taking place between the proteins of CLCuKoV-Bu and ToLCNDV on the basis of the speculations that this coinfection can lead to a third epidemic of the CLCuD. VVIs were determined by taking into account the PPI between the proteins of the two viruses. The findings reveal domains of viral proteins, responsible for performing crucial roles in the replication and translocation in the viral life cycle interacting with each other. This interaction shows that the proteins of the two co-infecting viruses may have an affinity, and bind easily to one another, altering the course of the disease in the host. The effect such interactions may have on the host can only be confirmed by in vivo analysis of protein complexes within the host plant The residues and domains highlighted in this study may provide a valuable insight into virus-virus interactions and can serve a lead in understanding the elusive VVIs. Strong interactions among the key domains of the proteins of the two co-infecting viruses suggest the interaction maybe antagonistic or a type of cross-protection or homologous interference, however such claims can only be proved by carrying out further experimentation.

Supplemental Information

Supplemental Information 1 Structure Assessment of the models generated for the proteins of CLCuKoV-Bu and ToLCNDV

Click here for additional data file.

Supplemental Information 2 Active and Passive Residues of the proteins of CLCuKoV-Bu and ToLCNDV

Click here for additional data file.

Supplemental Information 3 Comparison on interactions taking place between Intra-viral and inter-viral interaction between the two viruses

Click here for additional data file.

Supplemental Information 4 HADDOCK provided scores for all the interactions observed between the proteins of CLCuKoV-Bu and ToLCNDV

Click here for additional data file.

Supplemental Information 5 Binding Affinity ΔG (kcal mol-1) and the Dissociation Constant Kd (M) at 25.0 for the interactions observed between the proteins of CLCuKoV-Bu and ToLCNDV calculated through PRODIGY server

Click here for additional data file.

Supplemental Information 6 Number of Interfacial contacts (ICs) and Non Interacting Surface per property (NIS) present among the docked complexes

Click here for additional data file.

The present study was completed in 2020, as a joint collaboration between Atta ur Rahman School of Applied Biosciences (ASAB) and Research Centre for Modelling and Simulation (RCMS) at the National University of Sciences and Technology.

Additional Information and Declarations

Competing Interests

Author Contributions

Data Availability

The authors declare there are no competing interests.

Nida Fatima Ali performed the experiments, analyzed the data, prepared figures and/or tables, and approved the final draft.

Rehan Zafar Paracha conceived and designed the experiments, authored or reviewed drafts of the paper, and approved the final draft.

Muhammad Tahir conceived and designed the experiments, authored or reviewed drafts of the paper, provided Theoretical Framework for the study, and approved the final draft.

The following information was supplied regarding data availability:

Raw data is available in the Supplemental Files.

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
