# Peer review of "In silico evaluation of molecular virus–virus interactions taking place between Cotton leaf curl Kokhran virus- Burewala strain and Tomato leaf curl New Delhi virus"

_PeerJ, doi:10.7717/peerj.12018_

## Round 0.1 · original submission · Major Revisions

Please kindly attend to the reviewers’ comments very diligently and thoroughly. Looking forward to your revised manuscript. Thank you very much.

Reviewer 1 ·

Basic reporting

The written English language is OK but could be improved. The word "the" is frequently used and misused throughout the manuscript. Additionally, the authors used non-scientific terms (possibly to avoid plagiarism), which should be corrected. Need to be succinct and specific.

At several places, irrelevant references are cited (see the comments).

Figures 1 and 2 – Authors failed to produce the exact genome organization of begomoviruses, esp the C4 and TrAP (C2) ORFs. The C2 is truncated (~35 aa) in CLCuKoV-Bur only but not in ToLCNDV, but the authors illustrated the same for both the begomoviruses. Similarly, the size of C4 ORF is usually variable, on some viruses it completely localized within the Rep and sometimes does not. So, the sizes of the proteins depict the presence of anomalies. If authors used some particular sequence of a bipartite begomovirus then they should mention the accession number here (or refer to a suitable table). In addition, the sizes of DNA A and DNA B are not the same. These anomalies raised serious concerns over the sizes of the used proteins, their structures, and protein-protein interaction presented in the study.

Experimental design

The designed study is within the aims and scope. But the authors superficially discussed their results (see the comments).

The demonstrated In silico analyses are performed to a substantially good level. However, these findings should be supported with some wet lab work (see the comments).

Validity of the findings

The performed in silico work is good enough to decipher the molecular interactions between the subjected proteins.

The authors concluded “the antagonistic behavior between the investigated viruses”. Although, this is far from the conclusion – as the opposite is also possible. However, this notion should be addressed by co-inoculating these two viruses, in all possible combinations, in host/model plants (the same is mentioned in comments too)

Additional comments

1. It is a well-established fact that viruses interact (in a synergistic or antagonistic manner) during co-infections. The results obtained here indicate that the majority, but not all, of the proteins interact with the same protein from another virus, such as ToLCNDV-Rep and CLCuKoV-Bur-Rep. Again, it is well established that the majority of these proteins form oligomers as a result of their interacting residues – thus, these interactions are simply a result of those specific domains.
2. The produced title is not a well fit. The word in silico must be included.
3. Overall, the introduction section lacks a sequential presentation and did not provide vital information. Based on the results - authors should demonstrate “how Rep performs it function, esp the hypotheses demonstrated by Nawaz ul Rahman et al., 2009. Additionally, the study demonstrated by Akbar et al., 2016 could be discussed.
4. Scope exists to improve the ‘Discussion’ section. Specifically, the work presented by Zaidi et al., 2017 and Iqbal et al., 2017 may be beneficial in discussing the region's current situation in light of the findings of the presented study.
5. The authors concluded “the antagonistic behavior between the investigated viruses”. Although, this is far from the conclusion – as the opposite is also possible. However, this notion should be addressed by co-inoculating these two viruses, in all possible combinations, in host/model plants.

Minor Comments
L 80: italicize the genus name.
L 81: not the exact reference.
L87: not an exact reference.
L91: As authors are using a bipartite begomovirus, they may use (A)V2 rather than just V2. Do the same for other ORFs too, wherever is applicable.
L94: replace “anti virion sense strand” with a “complementary sense strand.”
L95: supplemented? Not a precise term. Need substantial English improvement.
L96: authors should mention the family and genus of beta- and alphasatellites.
L100-102: ambiguous sentence – rewrite it.
L140 & 145: authors must learn to write the name of begomoviruses according to the recent ICTV taxonomy.
L153: cotton samples?
L178-179: mention accession numbers.
L239: Reword the sentence to “Protein-protein Interactions (PPIs) of CLCuKoV-Bur and ToLCNDV encoded proteins…..”
Table 1: Mention the size of the proteins. In the first column of the table mention Virus/satellite
L315, 319, 322, 327, & 333: This looks somewhat more similar to the M&M section rather than the result section. Better omit it or describe it as a result section.
L323: replace the word “depicted” with either revealed or showed.
L323: percentage identity is a meaningless term. Better change it to “Percentage sequence identity”.
L375: replace “associative” with “associated”.
L447: Sentence needs a full stop.
L444 & 452: results of these parameters are not mentioned.
L473: looks quite odd using the term “interacted” here.
L481: italicize “in vivo”
L482: formatting needed.
L510: Authors failed to accredit the exact reference here. Zaidi et al. 2016 is not an exact reference.
L514: Should discuss the work performed by Akbar et al., 2016 here.
L517: Self-isolation? Please elaborate this phenomenon.
L533: omit parenthesis.
L541: Substitute navigation with a precise scientific word.

Reviewer 2 ·

Basic reporting

In general the article is well written with professional English used. Further editing to ensure minor typos are fixed and correct words are used would be beneficial, in particular spaces between numbers and symbols (e.g 3.0 kb not 3.0kb) and conversely not consequently. Some sections could be combined to reduce wordiness (e.g RMSD and RMSF lines 444 to 459). There are too many figures and tables in the article. These could be reduced to only show differences, for example, figures 8-11. Alternatively some could be put in supplementary data.

The study is very interesting and the choice to predict protein interactions using models is worthy of investigation. The conclusions drawn, however, are not supported by the evidence presented in the article. The reference to ToLCNDV being similar to CLCuKoV-Bur demonstrates a lacking in virology input to the article. They are both begomovirus species, but one is a bipartite and the other a monopartite which are quite different. The explanation of results in relation to virus epidemiology are not clear. Further detail is provided below.

Experimental design

The use of in silico modelling of protein interactions is very interesting and a useful technique to investigate virus-virus interactions.

The meaningfulness of the results presented would be improved if data was presented on homologous protein interactions. For example, what is the strength of the interaction of the CLCuKoV-Bur Rep with its own MP, NSP and TrAP and similarly for the ToLCNDV proteins with themselves. This would allow comparison of the strength of the interactions between proteins of the heterologous viruses to their own predicted interactions. Having this will improve inferences made on the likelihood the interactions would affect virus replication or movement etc. Without this its difficult to put the results in perspective. As such the experimental design is considered flawed in relation to linking the results with functional evidence of virus replication and/or in planta movement.

Additionally, a measure of functionality to support the in silico predictions would be beneficial. This could be through mutational knockouts of infectious virus clones, particularly using the identified amino acid residues predicted to be involved in the interaction. The infectious clones could also be used to evaluate whether the interactions are antagonistic through measure of virus titre in active infections by qPCR.

Validity of the findings

The conclusions made about protein interactions are well presented in relation to the likely strength of the interaction using multiple complementary methods. The strength of the interaction is further supported by detailed analyses of amino acid residues involved in the interaction. This analyses would benefit from the addition of the homologous virus protein interactions suggested above. It provides an internal control for the models used to measure protein interactions and puts the results of heterologous virus protein interactions into perspective.

The speculation on the interactions affecting virus epidemiology is logical. The conclusion, however, doesn't fit the data provided. As there is no functional evidence to show if a dual virus infection enhances or reduces virus replication or spread of one of the viruses, the speculation must cover all contingencies. Instead the article is indicating the interactions was antagonistic due to the spread and virulence of one virus species being reduced. There is no evidence in the article that supports that conclusion.

Additional comments

The study is very interesting and the use of in silico predictions for protein interactions is highly valuable as a first step in virus-virus interaction studies. Further data analyses is suggested, however, before the article is progressed for publication. This is to ensure the overall conclusions are valid in relation to predictions on virus epidemiology. For stronger investigation of the virus-virus interactions, functional experimental work is needed.

Reviewer 3 ·

Basic reporting

I have evaluated the manuscript. manuscript is interesting and written clearly. Figures, tables and their captions are self-explanatory. But there are some short coming especially in introduction and discussion section with reference to literature. Although historic data is very important, but authors should use recent literature to present the current state of problem. Moreover, results must be discussed with reference to both old and new literature

Experimental design

Experimental design and statistics are presented in a very good manner and in sufficient details and meet the journal standards

Authors should present, when and where this study was conducted? i.e. lab, years etc.

Validity of the findings

results are presented in a sufficient details, but authors are suggested to add some quantified values in the results section.

Although results are discussed and justified, but authors are suggested to use recent literature to justify the results to present current state of knowledge

In conclusion section, authors must present scope of the study

Additional comments

Please address the above stated suggestions to improve the manuscript

---

## Round 0.2 · Minor Revisions

Please kindly attend to the reviewer comments.
Thank you

Reviewer 2 ·

Basic reporting

Further editing of the manuscript is needed, particularly in the results and discussion sections. Comments are provided on the manuscript. Essentially some results are presented in the discussion section without appropriate description in the results section.

Experimental design

Authors addressed previous comments very well. No further comments for this section.

Validity of the findings

As mentioned in point 1 - further editing of the results and discussion section is needed to clearly identify and describe the findings of the study. Comments are on the manuscript.

Additional comments

Congratulations on an interesting study, I look forward to reading the next manuscript describing the biological data further investigating this virus interaction.

Annotated reviews are not available for download in order to protect the identity of reviewers who chose to remain anonymous.

---

## Round 0.3 · accepted · Accept

Thank you for addressing all the comments raised by reviewers. The peer-review process is very useful, authors benefited from it, and improved the quality of their work. The revised manuscript is now acceptable for publication. Thank you for finding PeerJ as your journal of choice. Looking forward to your future scholarly contributions. Congratulations and very best regards.